# A Systematic Review of Location Data for Depression Prediction

**DOI:** 10.3390/ijerph20115984

**Published:** 2023-05-29

**Authors:** Jaeeun Shin, Sung Man Bae

**Affiliations:** 1Department of psychology, Chung-Ang University, Seoul 06974, Republic of Korea; rheai@hanmail.net; 2Department of Psychology and Psychotherapy, Dankook University, Cheonan 31116, Republic of Korea

**Keywords:** depression, location, smartphone, geographical positioning system (GPS)

## Abstract

Depression contributes to a wide range of maladjustment problems. With the development of technology, objective measurement for behavior and functional indicators of depression has become possible through the passive sensing technology of digital devices. Focusing on location data, we systematically reviewed the relationship between depression and location data. We searched Scopus, PubMed, and Web of Science databases by combining terms related to passive sensing and location data with depression. Thirty-one studies were included in this review. Location data demonstrated promising predictive power for depression. Studies examining the relationship between individual location data variables and depression, homestay, entropy, and the normalized entropy variable of entropy dimension showed the most consistent and significant correlations. Furthermore, variables of distance, irregularity, and location showed significant associations in some studies. However, semantic location showed inconsistent results. This suggests that the process of geographical movement is more related to mood changes than to semantic location. Future research must converge across studies on location-data measurement methods.

## 1. Introduction

The number of people reporting mental health problems increased during the COVID-19 pandemic; in particular, the incidence of depression has increased by 25% since the pandemic [1]. Depression easily recurs and is one of the factors that cause individuals to develop a wide range of maladjustment problems. Depression is often accompanied by physical illness, loss of occupational functioning, and low quality-of-life problems [2]. Additionally, it is associated with high suicide rates [3,4]. Therefore, early detection of and intervention for depression can prevent its recurrence from becoming chronic [5] and can help prevent various maladaptive problems that individuals may experience due to depression.

The primary classification for diagnosing depression is the American Psychiatric Association’s The Diagnostic and Statistical Manual of Mental Disorders 5th Edition (DSM-5) [6]. According to the DSM-5, depressed mood and loss of pleasure or interest lasting for at least two weeks are the main core symptoms of depression. Depression diagnosis includes various symptoms such as weight loss or weight gain, sleep disturbances (insomnia or hypersomnia), psychomotor agitation or retardation, fatigue, feelings of worthlessness or excessive inappropriate guilt, cognitive difficulties, and suicidal thoughts and/or behavior. Depression is also accompanied by significant distress and functional impairment.

Existing methods for measuring and diagnosing depressive symptoms have mostly relied on self-report questionnaire ratings or clinician ratings based on individual subjective reports. However, these methods are affected by recall bias [7]. Additionally, because symptoms are subjectively experienced, self-report methods assessed by the patient’s experiences and perceptions of the symptoms may not accurately capture the impact of the symptoms on the patient’s behavior and function. The self-report questionnaire would be appropriate to measure symptoms related to subjective perceptions, such as depressed mood, feelings of worthlessness or excessive inappropriate guilt, and suicidal thoughts. However, behavioral symptoms of depression, such as sleep disturbances (insomnia or hypersomnia), psychomotor agitation or retardation, and fatigue, and functional effects of depression (e.g., reduced social interaction) can be more accurately measured through objective observation of behavior than through questionnaires [8]. The development of digital technology and the spread of personal digital devices, including smartphones, has made it possible to collect objective data that captures the features of depressive symptoms through passive sensing technology using digital devices. In a systematic review study conducted by Zarate et al. [9], digital data that capture symptoms of depression were defined as a digital phenotype (DP) of depression; DP used in an ecological and granular manner made it possible to objectively observe individual behavioral features of depression. In other words, measuring depression with DP compensates for the bias of the subjective self-report measurement method and enables more ecological and objective measurement.

The major challenges of DP research are to identify diverse digital data that enable objective observation and determine their relative usefulness for predicting and diagnosing depression. The variability in aspects of sleep, degree of social engagement, psychomotor speed, and activity, as reflected by SMS texts, calls, GPS, actigraphy, and screen time, is associated with symptoms of depression such as depressed mood, difficulty in concentration, diminished motivation, and reduced social engagement [10,11,12,13].

Location data explain the movement patterns and activity of participants and can predict the severity of depression with high accuracy [14]. In a systematic review of depression DP conducted by Rohani et al. [15], measurements such as SMS text messages and cell phone use frequency showed inconsistent results among studies in correlation with depression [16,17,18,19] and mobile phone screen transition time [19,20]; screen variables, such as active frequency [21], showed unclear correlations with depression. Meanwhile, homestay [20,22,23,24], location clusters [19,20,25,26], and entropy [19,20,24] derived from user location data consistently showed a significant correlation with depression and a high statistical significance across several studies.

Location data, primarily measured using GPS data, were captured as a series of time-stamped longitudes and latitudes. Because the raw GPS format is difficult to interpret and analyze, formulas are used to convert individual location-based behavioral feature variables that can be interpreted to reflect individual movement patterns [20].

Müller et al. [27] classified the various variables of location data used in previous studies into four features: distance, entropy, irregularity, and location. Distance features include variables derived from the distances between GPS coordinates. Entropy features refer to the variability of time spent in various places rather than a simple concept of distance. High entropy refers to an even allocation of time to visit different places and engage in different activities. Irregularity features describe the irregularity of an individual’s daily movement patterns. Location features include the number and cluster of locations visited by an individual.

Rohani et al. [15] noted that these location data were promising features for measuring and predicting depression and that methods for extracting features of physical activity, social activity, and mobility based on GPS data should be standardized across studies. Before deciding on a standardized data extraction method, it is necessary to synthesize various measurement methods for location data and identify the location-based variables that are effective in measuring and predicting depression.

Therefore, in this systematic review, digital phenotype studies on depression, which have been rapidly accumulating since 2010, were synthesized. Through this, we tried to identify the characteristics of location data that have been reported to show a consistently high correlation with depression and a relatively accurate predictive power for depressive symptoms. The purpose of the present review is to determine the usefulness of location data and the relative importance of individual location variables that can be used to measure and predict depressive symptoms. Through this work, standardization and active utilization of location data can be promoted in future DP research on the diagnosis and prediction of depression. In addition, it can be used to discriminate location variables that should be treated as relatively important.

## 2. Methods

This study was designed in compliance with the Preferred Reporting Items for Systematic Reviews and Meta-Analyses (PRISMA) statement [28] and reports the systematic review results.

### 2.1. Search Strategy

The search was performed using Scopus, PubMed, and Web of Science electronic databases. Papers in English were searched, and terms such as location, GPS, passive sensing, and depression were combined in the search using keywords and abstracts. We limited the studies to those published after 2010. The search was performed on 18 October 2022.

### 2.2. Selection of Relevant Studies

Studies for inclusion in this review were selected based on the following criteria: In terms of participant type, we included studies that examined participants with a diagnosis of depressive disorder or depressive symptoms and studies that investigated participants without a diagnosis of depressive disorder but analyzed participants to identify a depressive disorder or depressive symptoms.

Additionally, we included studies reporting non-invasive monitoring using commercially available wearable sensors (e.g., smart watches, mobile phones). Studies describing internet-based interventions without a sensor-based mobile app component or self-reported interventions based on questionnaires were excluded.

For result type, we included studies reporting correlating mental health conditions (e.g., depression) with location sensor-based data and excluded studies that provided a description of a mobile app but did not demonstrate statistical results.

Considering the trends in technological evolution, studies published between January 2010 and October 2022 were included, and full-text papers in English were included.

This process aimed to obtain an integrated understanding of appropriate studies related to the research topic; there were no restrictions on the study design, study quality, or sample size. Of 911 studies, Endnote was used to delete duplicates (n = 333). Two reviewers individually reviewed all titles, abstracts, and keywords of 578 papers to identify potentially relevant articles; 119 studies were eventually screened. Afterwards, we retrieved the full text of 119 studies selected as potentially relevant to the research topic, and the same reviewers assessed for relevance. Disagreements were resolved by discussion. In total, 31 studies were included in this review.

### 2.3. Data Extraction and Synthesis

A data abstraction form was developed, and the final form included study characteristics, target population (depressive patient, normal population, students, etc.), location data measurement technology (GPS, Wi-Fi, GSM cellular network, etc.), sampling frequency, outcomes (results between clinical state [depressive symptoms] and location data), and data measured in addition to location data.

Two researchers independently extracted data from the included studies; discrepancies between researchers were resolved through discussion. We performed a narrative synthesis on the extracted data, focusing on three main areas: study characteristics, technical and methodological aspects, and associations with depression and location data.

## 3. Results

Figure 1 presents a flowchart of our systematic review. In total, 578 individual studies were screened by their titles and abstracts. Through a full-text analysis of 119 studies judged potentially relevant, 88 were excluded according to the exclusion criteria. Finally, 31 studies were included in the present review.

### 3.1. Study Characteristics

Table 1 shows the general characteristics of unique studies.

Of the 31 studies, 21 (68%) [14,27,35,36,37,38,39,40,41,42,43,44,45,46,47,48,49,50,51,52,53] were conducted after 2019.

The largest number of location data studies was conducted in the United States (20, 65%) [10,19,20,22,27,30,31,32,33,34,35,37,39,41,44,46,47,48,50,51] followed by six (19%) [29,36,45,49,52,53] in Europe and five (16%) [14,38,40,42,43] in other regions.

Of the 31 studies, 10 (32%) [10,20,22,27,31,32,35,37,39,51] were conducted with college students, 8 (26%) [14,38,40,41,42,47,52,53] targeted the group diagnosed with depression, and 7 (23%) [19,29,30,37,43,45,46] worked with the normal group. Three studies (10%) [36,49,50] involved both depressed and normal groups. The remaining three studies (10%) [33,44,48] included groups with coexisting depression and other psychological disorders.

In terms of sample size, 15 studies (48%) [10,20,22,31,32,33,35,36,41,43,45,47,48,49,50] included more than 30 to less than 100 participants, followed by 7 studies (23%) [14,29,37,38,39,40,51] between 100 and 200 participants, and 8 studies (26%) [27,30,34,42,44,46,52,53] with more than 200 participants. In terms of duration, 11 studies (35%) [10,14,20,32,33,34,37,38,43,48,50] were conducted between 8 and 12 weeks, and 8 studies (26%) [35,39,42,44,47,51,52,53] in more than 12 weeks. Five studies (16%) [29,30,36,40,45] were conducted between 4 and 8 weeks, and seven studies (23%) [19,22,27,31,41,46,49] in less than 4 weeks.

### 3.2. Technological and Methodological Characteristics

Technological and methodological characteristics are shown in Table 2. Of the 31 studies, 19 (61%) [14,19,20,22,27,31,34,36,37,38,40,42,43,44,45,46,47,49,50] used GPS alone for location data detection, and 5 (16%) [10,29,32,41,51] used a combination of GPS and Wi-Fi. Two studies (7%) [30,35] used GPS, Wi-Fi, or accelerometers.

Each study used a combination of GPS and other technologies, namely, GPS and GSM cellular network [53]; GPS and Accelerometer [48]; GPS, Wi-Fi, and cell tower [39]; and GPS, mobile phone tower triangulation, and Wi-Fi network [33]. Another study utilized location-based data obtained using Bluetooth alone [52].

For location data detection, two (6%) studies [22,31] used sampling in less than 5 min, six (19%) [19,20,30,36,44,50] in the 5 min interval, five (16%) [10,32,39,47,53] in the 10 min interval, and three (10%) [29,37,45] in the 15 min interval. Three studies (10%) [41,43,52] were sampled at intervals of more than 15 min.

One study [46] reported that the app registered location changes. In addition, a variable time sampling range of 3, 4, 5, and 8 min [38] or 5 to 10 min intervals was used [42].

In a study by Muller et al. [27], the location was captured whenever the participant moved. Studies using a combination of GPS, accelerometer, and Wi-Fi differentiated the sampling time unit according to the characteristics of sensing technologies [35,48].

Twenty-two (71%) studies [10,14,19,29,30,31,32,33,34,36,39,40,41,42,43,44,45,46,47,48,49,50] used other passive sensing data in addition to location data and included them in the analysis.

### 3.3. Association with Depression

Among 31 studies, 12 [14,29,30,39,40,41,44,46,47,48,50,51] studies predicted the depression state by integrating location data or combining location data with other passive sensing data. Sixteen studies [10,19,20,22,27,31,32,33,34,37,38,42,45,50,52,53] presented the relationship between depression and individual location-based data. Three studies [35,36,43] presented individual associations of depression with location data and combined the results with other additional sensing data.

#### 3.3.1. Prediction of Depressive Symptoms

BDI-II: Beck Depression Inventory-II; Depression Scale of DASS: the Depression, Anxiety and Stress Scale; SCID: Structured Clinical Interview; PHQ-8,9: Patient Health Questionnaire-8,9; MADRS: Montgomery-Åsberg Depression Rating Scale. For depression measurement, the Beck Depression Inventory-II (BDI-II) [54], the Center for Epidemiologic Studies Depression Scale-10 (CES-D10) [55], depression subscale from the Depression, Anxiety, and Stress Scales (DASS) [56], the Montgomery-Åsberg Depression Rating Scale (MADRS) [57], the Patient Health Questionnaire (PHQ)-2 [58], 8 [59], 9 [60], the Structured Clinical Interview for Mental Disorders (SCID) [61], and Depression surveys according to the International Disease Classification (ICD-10) [46] were used. Among them, PHQ was most frequently used by 21 studies [10,14,19,20,29,30,32,34,35,37,38,40,42,43,44,47,48,50,51,52,53].

The results of studies that predicted depressive states by integrating location data with other sensor data showed that the predictive power of the Area Under Curve (AUC) was 56 to 80.71%. The accuracy ranged from 59.26 to 87.2%. The F1 value was 62–91% (see Table 3). Most studies, except for two, used a combination of location data and other passive sensing data, such as sleep duration, physical activity, and mobile phone use traits, to predict depressive symptoms.

In a study by Chikersal et al. [39], separate feature sets for behavioral features related to Bluetooth, calls, campus maps, phone usage, sleep, and steps were extracted; the individual depression predictive power for the feature set was evaluated. It was reported that the F1 score was 0.62 for the predictive power of depression for the location feature. Meyerhoff et al. [44] reported repeated-measures correlation coefficients between four clusters (location, time, transition, and semantic location) of location-based sensing data and changes in depressive symptoms. However, since these two studies did not report the predictive power of individual variables included in the location data, there is a limit to confirming which variable has relative importance among various location-based variables in the location data.

#### 3.3.2. Correlations with Depressive Symptoms

It is necessary to examine studies that present the relationship between depression and each variable of the location data. A total of 57 types of location-based variables were collected from the 31 studies included in this review. Among them, 16 studies investigated individual associations between each variable and depression. On exploring the studies, it was confirmed that location-based variables were significantly negatively or positively correlated with depressive symptoms. The direction of the positive or negative relationship between depression and location-based variables is indicated by “+” or “−”in Table 4.

Based on Müller et al.’s [27] classification of four broad types and factor analysis of latent behavioral dimensions, individual location-based variables were classified into dimensions of distance, entropy, location, and irregularity. In addition, in this study, semantic location and Bluetooth-based characteristics were classified as separate dimensions (see Table 4).

Among each dimension, variables belonging to the entropy dimension showed the most significant correlations in several studies. Homestays reported in eight studies [20,22,32,35,37,42,43,53], entropy in five studies [20,35,36,45,53], and normalized entropy in four studies [19,20,35,43] showed significant correlations with depression-related variables.

Next, in the distance dimension, the location variance variable showed a significant negative correlation with depression in six studies [19,20,35,43,45,50]. The total distance variable reported a significant negative correlation in three studies [10,34,50] and the mobility radius in one study [34].

In the case of the location dimension, the number of cluster variables reported a significant correlation in two studies [20,50]. The number of significant places [36], mean length of stay at clusters [36], and location count [53] each reported a significant correlation in one study.

Regarding the irregularity dimension, circadian movement reported a significant relationship in two studies [19,20]. Another study reported that the trait set combining the variables of the irregularity dimension showed a significant negative correlation with depression [27].

For semantic location, Boukhechba et al. [31] suggested a differential correlation with depressive symptoms depending on the type of visit to semantic places, such as home, leisure activities, and other people’s homes.

Regarding the Bluetooth-derived features, because only one study was conducted using Bluetooth [52], the relative importance of the variable could not be compared; however, three variables had significant negative correlations with depression.

## 4. Discussion

### 4.1. Summary of the Main Findings

In this study, a systematic review was conducted on the location data used to predict depression symptoms and to search for correlations with depression symptoms. In total, 31 studies were included in the narrative synthesis. Although the range of the search spanned from 2010, the included studies were mostly published after 2019, demonstrating the novelty of the research field. The sample sizes ranged from less than 30 to more than 200. There were 17 studies targeting the general group, including university students, and 14 studies targeting the clinical group, including the group with diagnosed depression. The duration of the studies also varied from less than 4 weeks to more than 12 weeks. For collecting location data, GPS was included in 30 of 31 studies, and Bluetooth was used alone in one study. GPS is most often used to collect location data, but 11 studies used a combination of GPS and other sensors (i.e., Wi-Fi, cell tower, Accelerometer). Yue et al. [35] compared using GPS alone and data fusion using GPS and Wi-Fi; the correlation between depression and location data was improved after data fusion. Further investigation is needed to determine which sensor combination is more helpful in predicting depression.

In addition to location data, studies that performed depression prediction, including behavioral feature variables using other passively sensed data, suggested the predictive power and usefulness of location data in predicting depressive symptoms (see Table 3). A study that systematically reviewed the usefulness of various passive sensor data in the detection of depressive and bipolar disorders [62] also emphasized the usefulness of location data.

According to dimension classification [27], many feature variables included in location data can be classified into four dimensions: entropy, distance, location, and irregularity. Variables such as entropy, normalized entropy, and homestay, which belong to the entropy dimension, have shown consistent and strong associations with depressive symptoms in several studies. The entropy variable is a measure of how evenly participants spend time across clusters of places. Low entropy means that the time spent in the visited places is not evenly distributed. Entropy is significantly negatively correlated with depression [20,35,36,45,53].

A study by Asare et al. [36] showed statistically significant lower mobility when comparing depressed and non-depressed groups; the depressed group tended to have lower entropy. Normalized entropy by the number of places visited showed a significant correlation with depression [19,20,35,43]. In a study by Saeb et al. [17], normalized entropy classified participants with depressive symptoms (Patient Health Questonnaire-9 (PHQ-9) score ≥ 5) and those without (PHQ-9 score < 5) with 86.5% accuracy.

Homestay refers to the time spent staying at home, where one spends the most time. The homestay variable is highly related to the entropy variable because it measures the amount of movement through space in various ways. A high homestay level reflects a low level of entropy. The homestay variable was also used as a single variable for predicting depression in some studies [22,42], and it showed consistent results in most studies [20,22,32,35,37,42,43,53]. Depression levels tended to increase as people spent more time at home. In a study that conducted a within-individual comparison, negative emotions increased as the time spent at home increased, and positive emotions increased as the time spent at home decreased [22].

A study of major depressive disorder (MDD) among patients of different ages and in different countries also verified the relationship between homestay and MDD symptom severity through a regression model; consequently, higher MDD symptom severity was related to long-term homestay [42]. This relationship was mainly confirmed in weekday data, and age and occupational status significantly modulated homestay [42]. Although the homestay level did not show a direct relationship with depression (no main effect on PHQ-9), it showed a significant main effect on daily stress levels. Moreover, an interactive effect between the homestay period and the period of not communicating with others was found. It had a significant effect on the changes in depression (PHQ-9) and loneliness (RULS) scores [32].

The distance dimension includes the total distance, maximum distance away from home, and location variance (the sum of variances of latitude and longitude values). Among these variables, location variance showed a significant correlation with depression in most studies within the distance dimension [19,20,35,43,45,50]. In a study by Moshe et al. [45], only location variance was found to be significant among GPS features in a regression analysis related to the predictive power for depression; high location variance predicted lower depression symptoms. People who exhibit more movement can be said to be less depressed. Furthermore, location variance, which reflects the variability of latitude and longitude that can be captured by GPS, can be said to reflect behavioral features related to people’s mobility. More intuitive concepts—such as total distance [10,34,50] or mobility radius [34]—within the distance dimension also showed a significant negative correlation with depression. In the study by Thakur et al. [50], location variance and total distance traveled showed significant negative correlations with the PHQ-9 score, and a t-test between the depressed and non-depressed groups showed a significant difference.

The location dimension is composed of variables that represent the number and clusters of places visited. The fewer places visited and the more time one spends in a single cluster of places rather than in a cluster of different types of places, the more likely one feels depressed [20,36,50,53].

Another feature related to location is irregularity, which goes beyond simply identifying the difference according to a large or small distance traveled. This allows us to capture the patterns of different daily activities and mobility. Circadian movement was found to be significantly correlated with depression [19,20,27]. Behavioral feature variables that reflect irregularities, such as tile sequence edit distance, location sequence edit distance, circadian movement, and routine index, were reduced in dimension to irregular features and were analyzed. People with movement patterns characterized by a high degree of irregularity tended to report fewer depressive symptoms and less loneliness than those with a low degree of irregularity.

However, when the location is classified according to its use and the relationship with mood symptoms is examined, each study shows different results. Boukhecba et al. [31] analyzed participant data using GPS and reported that semantic location labeled according to location type showed a significant correlation with depressive symptoms. Saeb et al. [30] found little significant relationship between the amount of time spent in a place and the symptoms of depression and anxiety reported by the participants, although GPS could be used to distinguish the type of place. Meyerhoff et al. [44] also categorized the places visited according to GPS coordinates but did not report a significant correlation with depression. Meyerhoff et al. [44] suggested that moving through a geographic space through movement patterns rather than the meaning of place is an indicator of mood symptom change. Inconsistent results regarding semantic location suggest that movement through geographic space, rather than the kind of places we visit per se, may be related to mood. In other words, mood states are more likely to be related to the process of reaching various places, including physical activity, rather than the semantic characteristics of the actual place itself.

Bluetooth-derived features were identified in one study [52], and it was difficult to grasp the relative importance of variables within Bluetooth-derived behavioral features. The corresponding study [52] extracted 49 Bluetooth features and divided them into 3 categories: second-order statistics, multiscale entropy (MSE), and frequency domain (FD). In the case of depression measured by the PHQ-8, the number of Bluetooth-connected devices and the periodicity of the connection decreased. Bluetooth-derived features are promising indicators that indirectly reflect features such as social connection and interaction, isolation, and mobility rather than directly reflecting the mobility of participants. In future studies, it will be necessary to actively explore the association and predictive power of Bluetooth-derived features in depression.

### 4.2. Implications and Limitations

In this study, we systematically reviewed studies that explored the relationship between location data and depression and reported the general characteristics of the studies as well as the predictive power and association of location data with depressive symptoms. Thus, it was possible to identify variables that consistently reported a correlation with depression among various location data. Each of the studies conducted thus far investigated various variables related to location-based behavioral features and showed differences in sampling methods. When predicting depression symptoms using location data, it is necessary to consider how to weigh the corresponding variables for predictive power when inputting them into a machine-learning model for variables that have been consistently presented as important for increasing accuracy.

As presented in Table 1 and Table 2, the number of samples, subjects, and study period differed significantly in each study; hence, there is a limit to generalizing the results. Considering these points, it is necessary to expand these findings to include more studies in the future. If agreement and standardization are made among researchers on the sampling period or the use of technology, it may be helpful to use location data as a more accurate indicator for predicting depression when using location-based behavioral features in the future. In addition, this review did not include the potential influence of factors that could affect the relationship between location data and depression, such as participants’ regions, poverty, and socioeconomic or occupational status [53]. A study by Laoiu et al. [42] found that job status significantly modulated homestay. In future studies, it is necessary to specify variables that moderate the relationship between location data and depression.

Additionally, studies dealing with depression and location data were reviewed in this study through a systematic review; however, important publications may have been excluded because the number of related studies continues to increase. In addition, as mentioned by Fraccaro et al. [63], because DP research is a new field, many uses of different terms exist that are not agreed upon between studies; therefore, a few studies may have been omitted due to differences in search keywords.

### 4.3. Future Directions and Perspectives

DP technology can have innovative functions in measuring and predicting mental health, but it also has the problem of invasion of privacy. A key consideration when implementing digital depression diagnosis and intervention technologies is that participants must consent to the utilization of their personal information [64]. To obtain consent from the participants and obtain accurate data, it is necessary to first confirm the benefits to the participants, such that the acquisition of the information will help improve depression [64]. In addition, it is necessary to prepare common guidelines for access rights and protection of DP data across the studies [8].

It is also necessary to go beyond simply detecting and predicting depressive symptoms by DP data. This can be done by utilizing digital phenotypes of depressive symptoms, such as location data, to immediately detect depressive symptoms and provide timely information on depression treatment. A just-in-time adaptive intervention [65] is a smartphone-based treatment provided when depressive mood is predicted by a machine-learning model that uses passively detected data without participants’ direct self-report. Previous studies reported an improvement in mental health outcomes with just-in-time adaptive interventions [29,66]. It is necessary to activate a service that immediately detects changes in participants’ depressive symptoms by applying a just-in-time adaptive intervention that immediately suggests a change in intervention method. This can help overcome the limitations of existing mental health services, such as the shortage of mental health professionals (doctors and psychologists), long waiting times, and high costs. To this end, location data that show a consistent relationship with depression need to be utilized more actively and efficiently.

## 5. Conclusions

The use of location data to predict depression is an active research area. Location data reflect changes in individual mood states and are consistently highly correlated with depression. Future studies must verify and integrate the appropriateness of the techniques used between studies.

## Figures and Tables

**Figure 1 ijerph-20-05984-f001:**
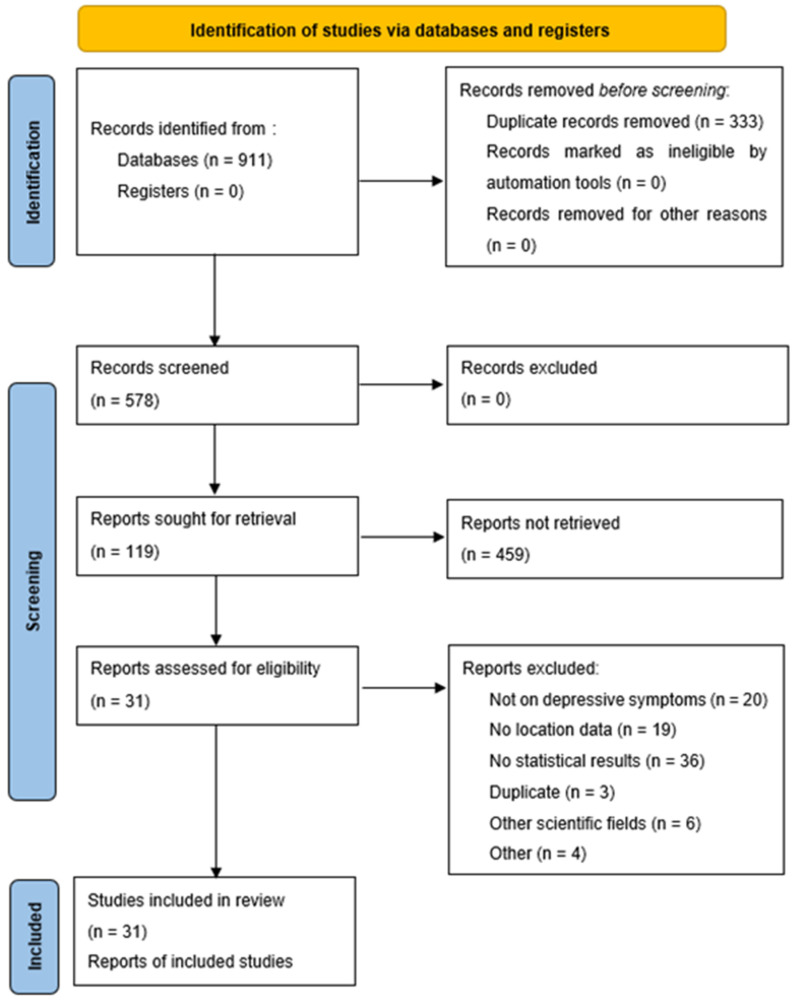
Flow diagram for screening and inclusion of relevant articles.

**Table 1 ijerph-20-05984-t001:** General characteristics of included studies (n = 31).

General Project Characteristic	Data *N* (%)	Reference(s)
Year of publication		
2014–2016	4 (13)	[10,19,20,29]
2017–2019	6 (20)	[22,30,31,32,33,34]
After 2019	21 (68)	[14,27,35,36,37,38,39,40,41,42,43,44,45,46,47,48,49,50,51,52,53]
Geographical locations		
Europe	6 (19)	[29,36,45,49,52,53]
United States	20 (65)	[10,19,20,22,27,30,31,32,33,34,35,37,39,41,44,46,47,48,50,51]
Other	5 (16)	[14,38,40,42,43]
Target Population		
Depressive disorder	8 (26)	[14,38,40,41,42,47,52,53]
Depressive disorder +Normal Group	3 (10)	[36,49,50]
Normal Group	7 (23)	[19,29,30,37,43,45,46]
University Students	10 (32)	[10,20,22,27,31,32,35,37,39,51]
other	3 (10)	[33,44,48]
Sample Size		
<30	1 (3)	[19]
30~100	15 (48)	[10,20,22,31,32,33,35,36,41,43,45,47,48,49,50]
100~200	7 (23)	[14,29,37,38,39,40,51]
More than 200	8 (26)	[27,30,34,42,44,46,52,53]
Study length		
<4 weeks	7 (23)	[19,22,27,31,41,46,49]
4–8 weeks	5 (16)	[29,30,36,40,45]
8–12 weeks	11 (35)	[10,14,20,32,33,34,37,38,43,48,50]
>12 weeks	8 (26)	[35,39,42,44,47,51,52,53]

Values are *n* (%).

**Table 2 ijerph-20-05984-t002:** Technological and methodological characteristics of included studies (*n* = 31).

Evaluation Characteristics	Data	Reference(s)
Technology to measure geolocation		
GPS	19 (61)	[14,19,20,22,27,31,34,36,37,38,40,42,43,44,45,46,47,49,50]
GPS + WiFi	5 (16)	[10,29,32,41,51]
GPS + WiFi + Accelerometer	2 (7)	[30,35]
GPS + GSM cellular network	1 (3)	[53]
GPS + Accelerometer	1 (3)	[48]
GPS + WiFi + Cell tower	1 (3)	[39]
GPS + mobile phone tower Triangulation + WiFi network locations	1 (3)	[33]
Bluetooth	1 (3)	[52]
Sample Frequency		
<5 min	2 (6)	[22,31]
5 min	6 (19)	[19,20,30,36,44,50]
10 min	5 (16)	[10,32,39,47,53]
15 min	3 (10)	[29,37,45]
>15 min	3 (10)	[41,43,52]
Others	8 (26)	[14,27,35,38,42,46,48,51]
Not reported	4 (13)	[33,34,40,49]
Additional sensor data collected?		
Yes	22 (71)	[10,14,19,29,30,31,32,33,34,36,39,40,41,42,43,44,45,46,47,48,49,50]
No	9 (29)	[20,22,27,35,37,38,51,52,53]

Values are *n* (%), GPS: global positioning system; GSM: Global system for Mobile communications.

**Table 3 ijerph-20-05984-t003:** Performance of depression detection using or including geolocation measures.

Reference	Included Features	Measure (F1, AUC, Accuracy etc.)	Results	Number of People	Time	Depression Symptoms Measurement
[36] Asare, K.O., et al. (2022)	Sleep Physical activity Phone usage GPS mobility	AUC	80.71	54	4 weeks	DASS Depression scale
[33] Place, S., et al. (2017)	sms.address.count travel.distance.sum		56	73	12 weeks	SCID (Depression A2—diminished interest or pleasure in all or most activities)
[36] Asare, K.O., et al. (2022)	Sleep Physical activity Phone usage GPS mobility	Accuracy	79.31	54	4 weeks	DASS Depression scale
[39] Chikersal, et al. (2021)	Location		69.5	105	16 weeks	BDI-II
[40] Hong, J., et al. (2022)	Location (location variance, entropy) Physical activity per day		59.26	106	4 weeks	PHQ-9
[43] Masud, M.T., et al. (2020)	Physical activity Movement (location) patterns		87.2	33	11 weeks	PHQ-9
[49] Sverdlov, O., et al. (2021)	Phone usage logs Geographic location data Wi-Fi sensor data		64	40	2 weeks	MADRS
[29] Wahle, F., et al. (2016)	Accelerometer Wi-Fi Global positioning systems		61.5	126	4 weeks	PHQ-9
[39] Chikersal, et al. (2021)	Location	F1	0.62	105	16 weeks	BDI-II
[14] McIntyre, R.S., et al. (2021)	Location (location variance, normalized entropy, number of clusters, total distance, and mean absolute deviation in distance) Daily call and SMS count		0.91	200	12 weeks	PHQ-9
[51] Ware, S., et al. (2020)	GPS data (Location variance, Time spent moving, Total distance, Average moving speed, Number of unique locations, Entropy, Normalized entropy, Time spent at home, Circadian Movement, Routine index)		MAX 0.83	79 (PHASE1)	8 months	PHQ-9
	Wi-Fi Data (Number of significant locations visited, Number of Entertainment, Sports, and Class Buildings visited, Average duration spent in Entertainment, Sports, Library, and Class buildings, Number of days visiting Entertainment, Sports, Library, and Class buildings)		MAX 0.86			
[35] Yue, C.Q., et al. (2018)	GPS + WIFI data fusion		0.76	79	8 months	PHQ-9
[41] Jacobson, et al. (2020)	(1) Direct location-based information: GPS coordinates (latitude, longitude), Location accuracy, Location speed, and Whether the location-based information was based on GPS or WiFi; (2) Location type based on the Google Places location type (e.g., University, gym, bar, church); (3) Local weather information: Temperature, Humidity, Precipitation, Light level, (4) Heart rate information: Average heart rate and Heart rate variability; (5) Outgoing phone calls.	Correlation with the observed scores from the models	r = 0.587, 95% CI [0.552, 0.621]	31	1 week	DASS Depression scale
[44] Meyerhoff, J., et al. (2021)	Locations (location cluster and location variance; represents the number and variability in locations visited)	Repeated measures correlation coefficient: Repeated measure correlations sensor and symptoms changes	−0.17 (*p* < 0.001)	223	16 weeks	PHQ-8
	Time (total entropy, normalized entropy, and circadian movement; represents the variability in time spent across locations)		−0.12 (*p* < 0.001)			
	Transitions (distance traveled and velocity; represents travel between locations)		−0.12 (*p* < 0.001)			
	Semantic location (Exercise location duration)		0.18 (*p* = 0.001)			

**Table 4 ijerph-20-05984-t004:** Relationship between depression and specific geolocation feature variables.

Dimension	Specific Feature	Direction of Relationship	Reference(s)
Entropy	Entropy	−	[20,35,36,45,53]
	Normalized Entropy	−	[19,20,35,43]
	Homestay	+	[20,22,32,35,37,42,43,53]
Distance	Total Distance	−	[10,34,50]
	Location Variance	−	[19,20,35,43,45,50]
	Mobility Radius	−	[34]
Location	Number of significant places	−	[36]
	Mean length stay at clusters	−	[36]
	Number of Clusters	−	[20,50]
	Location Count	−	[53]
Irregularity	Circadian Movement	−	[19,20]
	Irregularity features (Tiles sequence edit distance, Location sequence edit distance, Circadian movement, Routine index)	−	[27]
Semantic Location	Home	+	[31]
	Leisure	−	
	Service Location	+	
	Other’s home	−	
Bluetooth derived features	Second Order statistics	−	[52]
	Multiscale entropy (MSE)	−	
	Frequency domain	−	

## Data Availability

Not applicable.

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
