# Peer review of "A Systematic Review of Location Data for Depression Prediction"

_ijerph, 2023, doi:10.3390/ijerph20115984_

Round 1

Reviewer 1 Report

This review summarizes the relationship between depression and location data as measured through passive sensing technology. It was found that certain location-data variables, such as homestay, entropy, the normalized entropy variable, distance, irregularity, and location, all had significant correlations with depression while semantic location showed inconsistent results. It was concluded that geographical movement is more closely related to mood changes than semantic location and that location data shows potential for use in continuous monitoring and prediction of individuals' depressive symptoms. I read this paper with great interest and this paper is well written and organized. I have few comments and hope can improve the paper.

1.     This paper included studies had a wide variety of methodological approaches. This makes it difficult to compare the results directly between studies, as there are many potential sources of bias that can be introduced through the use of different methods. Furthermore, the studies included in the review had a wide range of populations, study designs, and outcome measures. This further reduces the ability to draw any generalizable conclusions from the review.

2.     The paper does not consider the potential impact of environmental factors on depression prediction, such as access to mental health services and the effects of living in an area with high levels of poverty or crime. Such factors could be important to consider when predicting depression, as they may influence individuals’ access to resources and support systems that could reduce the likelihood of depression.

3.     I would suggest that the study incorporates a broader range of research methods when selecting studies for inclusion in the review. For example, the review could include studies using qualitative methods (e.g. interviews, focus groups, etc.) in addition to quantitative studies as they could provide valuable information related to the connection between location data and depression. In addition, incorporating a wider range of data sources. Because there are other potential sources of location data that could be explored (e.g. GPS data, satellite imagery, etc.).

4.     In Table 1 and 2, I think the way of presentations is not informative. That is, readers may not be interested in the listing of reference numbers.  Is there a better way to present? Same concerns are in the context, e.g. Line 177 to Line 194.  

Reviewer 2 Report

The manuscript "A systematic review of location data for depression prediction" is an original and necessary review of publications, examining the relationship between subjective description of depression symptoms in questionnaires, and objective indices of depression, retrieved from digital devices. The manuscript, however, requires improvement in the introduction, results, and discussion sections:

1. Please describe the symptoms of depression and the definition of this disorder according to DSM-V and ICD-10. It must be clear what potential symptoms are checked with digital devices and which are not available with these technologies.

2. Please describe how depression symptoms are measured using self-reported questionnaires, including all these tools from Table 3, like DASS-21, SCID, MADRAS, BDI-II, PHQ-8 and PHQ-9 (and differences between PHQ 8 and 9). The references to the original resources about these questionnaires are also necessary. 

3. It is unclear what was correlated with DASS-21. The DASS 21 measures distress, including depression, anxiety, and stress, and only one subscale is strictly related to depression. It must be clearly stated whether the correlations in Table 3 is with the total DASS-21 or only depression. It is important to note that only depression subscales should be correlated. If Table 3 includes inappropriate data, the results should be corrected or deleted (all results with DASS-21) to avoid a fatal failure.

4. It is inappropriate to use the heading in Table 3 "Clinical ground truth", since questionnaires are self-report measures to screen depression symptoms, but it cannot be treated as a clinical diagnosis by a professional psychiatrist or psychologist. Please correct this mistake in the whole manuscript.

5. Table 4 should also be corrected if DASS-21 was included to "direction of relationship" between specific features of digital devices and depression.

6. Discussion must be improved, in accordance with the data corrected or deleted from the result section.

Reviewer 3 Report

Depression contributes to a wide range of maladjustment problems. In this paper, the authors made a systematically review to the relationship between depression and location data. The analysis angle of this paper is meaningful. However, the following problems should be clarified.

(1) The main contribution of this review and the main challenges for digital phenotype studies on depression or location data for depression prediction should be clearly provided in the introduction part.

(2) The overall structure of this paper should be improved. As a review paper, the characteristic and their contributions, the advantages/disadvantages of the main methods related to the research of location data for depression prediction should be summarized.

(3) It suggests more figures should be provided of describing the main features and the research results of the methods related to the research of location data for depression prediction.

(4) The challenges and the research future prospects related to the research of location data for depression prediction should be detailed summarized in the discussion part.

(5) The conclusion part and the abstract part should be more concise.

(6) The English writing of this paper should be improved.

Depression contributes to a wide range of maladjustment problems. In this paper, the authors made a systematically review to the relationship between depression and location data. The analysis angle of this paper is meaningful. However, the following problems should be clarified.

(1) The main contribution of this review and the main challenges for digital phenotype studies on depression or location data for depression prediction should be clearly provided in the introduction part.

(2) The overall structure of this paper should be improved. As a review paper, the characteristic and their contributions, the advantages/disadvantages of the main methods related to the research of location data for depression prediction should be summarized.

(3) It suggests more figures should be provided of describing the main features and the research results of the methods related to the research of location data for depression prediction.

(4) The challenges and the research future prospects related to the research of location data for depression prediction should be detailed summarized in the discussion part.

(5) The conclusion part and the abstract part should be more concise.

(6) The English writing of this paper should be improved.

Round 2

Reviewer 1 Report

No further comments from me. Good luck!

Reviewer 2 Report

The authors revised the manuscript appropriately, so I can accept the current version, except the heading "ground truth" in Table 3, which is completely unclear. I suggest "depression screening test" or "depression symptoms measurement". 

Reviewer 3 Report

The quality of the revised version is improved. The authors have addressed my concerns. However, please check the format and the English expression of this paper.

The quality of the revised version is improved. The authors have addressed my concerns. However, please check the format and the English expression of this paper.
